# Non-Synonymous Variants in Fat QTL Genes among High- and Low-Milk-Yielding Indigenous Breeds

**DOI:** 10.3390/ani13050884

**Published:** 2023-02-28

**Authors:** Neelam A. Topno, Veerbhan Kesarwani, Sandeep Kumar Kushwaha, Sarwar Azam, Mohammad Kadivella, Ravi Kumar Gandham, Subeer S. Majumdar

**Affiliations:** 1DBT—National Institute of Animal Biotechnology (NIAB), Hyderabad 500032, India; 2RCB—Regional Centre of Biotechnology, Delhi 121001, India; 3ICAR—Indian Veterinary Research Institute, Bareilly 243122, India

**Keywords:** milk fat, whole-genome sequencing, SNPs, genomic variation, variant calling, indigenous breeds

## Abstract

**Highlights:**

**What are the main findings?**
Differentially expressed milk fat QTL genes explored with whole genome se-quencing for variant analysis.Identified non-synonymous SNPs for hub and bottleneck QTL genes associated with milk fat traits.

**What is the implication of the main finding?**
Identified differential pattern(s) of SNPs in fat QTLs between high and low milk yield breeds.Impact of the identified SNP pattern(s) on milk fat traits can be further explored.

**Simple Summary:**

Milk fat is a crucial trait that varies significantly among cattle breeds and determines the milk quality and pricing value. Indigenous breeds have disparity in milk quantity and quality. Our study is one of a kind which helps to decipher the variations at the genetic level correlated with transcriptional level among high and low milk-yielding cattle breeds exploring the fat QTLs. We assessed and unveiled a few key differences between the high and low-milk-yield breeds.

**Abstract:**

The effect of breed on milk components—fat, protein, lactose, and water—has been observed to be significant. As fat is one of the major price-determining factors for milk, exploring the variations in fat QTLs across breeds would shed light on the variable fat content in their milk. Here, on whole-genome sequencing, 25 differentially expressed hub or bottleneck fat QTLs were explored for variations across indigenous breeds. Out of these, 20 genes were identified as having nonsynonymous substitutions. A fixed SNP pattern in high-milk-yielding breeds in comparison to low-milk-yielding breeds was identified in the genes *GHR, TLR4, LPIN1, CACNA1C, ZBTB16, ITGA1, ANK1,* and *NTG5E* and, vice versa, in the genes *MFGE8, FGF2, TLR4, LPIN1, NUP98, PTK2, ZTB16, DDIT3*, and *NT5E*. The identified SNPs were ratified by pyrosequencing to prove that key differences exist in fat QTLs between the high- and low-milk-yielding breeds.

## 1. Introduction

India has become the largest milk producer in the world [1]. Several schemes involving crossbreeding have been implemented to enhance milk production in the country. As a result, the number of crossbred cattle increased and contributed to around 28 percent of total milk production in India (ca. 188 million tons), surpassing the contribution of indigenous cattle [2]. However, indigenous cattle breeds are well known for their heat tolerance and disease resistance [3], and the crossbreds have been found to be susceptible to tropical diseases and harsh climatic conditions and require constant good management practices. To strike a balance between increasing demand for milk and the change in the environment due to global warming, exploring the genomic merit of indigenous cattle/breeds becomes even more important.

Milk being a polygenic trait with medium heritability, the majority of animal breeding research has centered on quantitative trait loci (QTLs) with moderate to large effects on milk production traits. The *DGAT1* on chromosome 14 [4,5,6], the growth hormone receptor (*GHR*) on chromosome 20 [7], and the *ABCG2* [8] or *SPP1* (Osteopontin) on chromosome 6 [9] are well-known QTL genes that have been fully characterized with a strong putative or well-confirmed causal mutation. The two QTLs *DGAT1* (K232A) and *ABCG2* (Y581S) in *Bos taurus* have been suggested to be associated with increased fat yield and fat and protein percent in milk with a decrease in milk yield [6,8,10,11,12,13]. The *GHR* mutation F279Y has been observed to have a significant effect on milk composition (fat and protein percentage) and milk yield [14]. The locus c.8514C > T in the intronic region of *SPP1* has also been found to have a significant effect on milk production and milk composition [15]. However, the *DGAT1* and *ABCG2* genes have been found to be fixed among Indian breeds Sahiwal, Rathi, Deoni, Tharparkar, Red Kandhari, and Punganur [16]. Currently, the animal QTL database (QTLdb) contains 1,93,216 QTLs for different bovine traits, out of which 83,458 QTLs have been reported for milk traits [17].

Milk is the primary source of nutrition for infants, as well as adults. Besides its nutritional value, it has a major role in imparting growth and immunity through intrinsic milk components such as growth factors, chemokines, anti-inflammatory molecules, antioxidants, prebiotics, and probiotics [8,9]. Milk has four major components, fat (3.6%), protein (3.2%), lactose (4.7%), and water (87%), along with other various kinds of minerals, enzymes, vitamins, and dissolved gases. Various research studies have shown that several factors, such as lactation stage, genetics, environmental factors, and diet management, influence milk quality. The variability of milk composition among popular dairy breeds Brown Swiss, Holstein Friesian, Jersey, Simmental, Grey Alpine, and Rendena under the same dairy management practices has been explored, and Holstein Friesian had higher milk yield with lower fat content (27.45 kg/d, 4.04%) [18], whereas Jersey had lower milk yield with relatively higher fat content (17.27 kg/d, 5.65%) [13]. A low fat percentage has also been reported for Ayrshire, Brown Swiss, Guernsey, Holstein Friesian, and Jersey breeds in the United States [19]. Furthermore, the effect of breed has been found to significantly influence the water (*p* ≤ 0.0001), protein (*p* ≤ 0.05), total solids (*p* ≤ 0.05), fat (*p* ≤ 0.05), milk urea nitrogen (*p* ≤ 0.001), and ash (*p* ≤ 0.0001) content of milk [20].

India, with a huge diversity of 50 cattle breeds, forms an ideal ground to study genetic variation at the genomic level vis-à-vis milk traits [21]. The cost of the milk world-over varies with the percentage of fat present in the milk. Exploring the variations in fat QTLs across breeds would shed light on the variable fat content in their milk. No such studies have been reported in the past for indigenous cattle breeds to evaluate the variation across indigenous breeds within the fat QTLs. Therefore, the objective of the present study was to explore genomic variation(s) within the fat QTLs that were identified to be differentially expressed in lactation across indigenous breeds, which were divided into high- (Sahiwal and Gir) and low-milk-yield (Gaolao, Deoni, Pulikulam, Hallikar, Dangi, and Amritmahal) groups.

## 2. Materials and Methods

### 2.1. Data Retrieval from the Public Repository

QTL genes associated with milk fat traits (milk fat content and percentage) and metabolism were extracted from the Animal QTLdb database [22], and the duplicates were removed. QTL genes (286 and 256 (total of 542)) were identified for milk fat yield and milk fat percentage, respectively, with 417 unique genes for both traits (Appendix A). Functional annotation of the 125 QTL genes commonly associated with milk fat yield and milk fat percentage was performed in g:Profiler [23] and ShinyGO [24] to identify the enriched biological processes. Protein interaction network analysis of the 417 genes was performed using the Search Tool for the Retrieval of Interacting Genes Search Tool for the Retrieval of Interacting Genes 11.0 (STRING 11.0) database at a confidence score value of 0.5 against model species *Bos taurus* [25]. The interaction network was imported into the Cytoscape 3.8.0 software (Institute for System Biology, CA, USA) for visualization. The hub and bottleneck genes were identified in the interaction network using the Cytohubba plugin of Cytoscape [26] considering the degree of association between the genes and by taking the bottleneck approach, which takes into account the top 20% of the degree of distribution of the proteins in the network [26]. A total of 74 QTL genes were identified to be hub/ bottleneck genes. The hub genes were the genes that had the highest degree of association, and the bottleneck genes were the key connectors having a high betweenness (measure the centrality of the nodes) among different clusters in protein interactions [27].

### 2.2. Bioinformatics Analysis of Milk Transcriptome

The publicly available milk transcriptome bioproject ID (PRJNA419906) was used to analyze the expression of QTL genes associated with milk fat traits. This bioproject was considered in this study as it has data generated from Jersey (a breed with high fat content ranging from 4.10–4.86%) and Kashmiri (a breed with low fat content ranging from 3.20–3.94%) [28]. The data were generated from mammary epithelial cells (MECs) collected on Day 15 (D15), D90, and D250 from six lactating cows (three Jersey and three Kashmiri cattle). These days represent early, mid-, and late lactation, respectively [28]. The data were downloaded from the Sequence Read Archive (SRA) of the NCBI database, and the fastq-dump program of SRAtoolkit [29] was used to extract the fastq reads. Quality assessment and control of RNA-seq data were performed through Fast QC Version 0.11.5 [30], MultiQC Version 1.8 [31], and trimmomatic Version 0.39 [32]. All the high-quality reads were mapped to the *Bos indicus* genome (GCF_003369695.1) using STAR Version 2.5.4b with the default parameters [33]. Gene expression was estimated using RSEM [34], and differential gene expression was performed through the DESeq2-R package [35]. The differentially expressed genes among the 74 fat QTL hub and bottleneck genes were identified. 

### 2.3. Breed Selection, Sampling, Genomic DNA Extraction, and Whole-Genome Sequencing

Indigenous cattle breeds for the proposed study were grouped into high- and low-milk-yield-breed groups. The high-milk-yield group (avg milk yield per day 8 kg) included Sahiwal (n = 4) and Gir (n = 4), and the low-milk-yield group (avg milk yield per day 2.5 kg) included 6 animals representing the breeds Gaolao, Deoni, Hallikar, Dangi, Pulikulam, and Amritmahal [36]. Animals of different breeds were considered in the study to have a true representation of both the high- and low-milk-yield groups. Genomic DNA (gDNA) was extracted from the blood samples of the animals from these breeds using the nucleospin blood L-kit (Macherey-Nagel), and the integrity of the genomic DNA was checked on agarose. After estimating the concentration of gDNA (Nanodrop2000, ThermoFischer Scientific), DNA libraries were prepared as per the manufacturer’s protocol (Illumina sequencing platform) for paired-end sequencing (2150 bp).

### 2.4. Bioinformatic Analysis (Variant Calling, SNP Annotation, and Functional Enrichment)

Sequencing data generated on an Illumina platform were pre-processed for quality assessment and improvement (base quality, nucleotide distribution, GC content, adaptor sequence, duplication, length distribution, etc.) by FastP [37]. All high-quality reads were mapped to the *Bos indicus* reference genome (Brahman-GCF_003369695.1) using BWA aligner [38]. Variant calling was performed from the aligned data using freebayes [39] and GATK [40]. For the GATK- and freebayes-generated vcf files, only SNPs were selected, leaving aside all indels and insertions. After freebayes variant calling, the low-quality variants were filtered by vcftools Version 1.10 [41] for Q > 20. In the GATK pipeline, the paired-end Illumina Hi-Seq raw reads for each individual were first converted into an unaligned bam; the illumina adapters were marked; the sam was converted back to FASTQ; the reads were mapped to the reference Brahman genome (GCF 003369695.1); the unaligned and mapped bams were combined. Finally, the duplicate marked clean bam was used to generated GVCF for each animal using GATK haplotypecaller. The GVCFs generated for all the animals were combined to call the variants using the genotypeGVCF module. The parameters for GATK VariantRecalibration to generate the VQSLOD score were QD < 2.0, MQRankSum < −8.5, ReadPosRankSum < −8.0, FS > 60.0, MQ < 40.0, SOR > 3.0, and DP 30x (depth or coverage). The final set of SNPs after recalibration by GATK included the selection of SNPs that passed. We further used the GATK-filtered set and the freebayes-filtered set to identify the common SNPs across these variant callers. From this vcf file, the SNPs in the differentially expressed hub and bottleneck genes (i.e., genes that are hub/bottleneck and are differentially expressed as identified in Section 2.2) were extracted using an in-house perl script. The non-synonymous SNPs (nsSNPs) in the coding regions of these were identified through the SnpEff tool [42].

### 2.5. SNP Validation through Real-Time Sequence-Based Pyrosequencing

Three nsSNPs that were found to be distinctly different between the high- and low-milk-yield groups were selected for validation and were genotyped in PyroMark Q48 (Qiagen) as per the manufacturer’s protocol. These nsSNPs were found in the differentially expressed hub and bottleneck genes *GHR, LPIN1*, and *TLR4*. GHR was one of the genes having the maximum number of interactions in the network. *LPIN1* had the maximum SNP count of 10, whereas *TLR4* was one of the top 10 highly upregulated genes. PCR and sequencing primers were designed using PyroMark Assay Design Software 2.0 (Qiagen). The PCR amplification was performed in a 20 µL reaction, with the thermal cycling conditions, which included an initial denaturation of 95 °C for 3 min followed by 40 cycles of 95 °C for 30 s, 65 °C for 30 s, 72 °C for 1 min, and a final extension of 72 °C for 10 min. Sequence analysis was performed by PyroMark Q48 Autoprep software Version 2.4.2 in SNP analysis assay mode for 14 animals.

The schematic representation of the study is given in Appendix A.

## 3. Results

### 3.1. Meta-Analysis of Cattle-Milk-Fat-Component-Associated QTLs and Related Genes

Animal QTLdb was used to extract fat-trait-associated QTL genes. In the QTLdb, after the removal of duplicate genes (Appendix A), 286 and 256 genes were found associated with milk fat yield and percentage, respectively, whereas 125 common genes were found between milk fat yield and percentage (Figure 1A). A total of 417 unique genes were found to be associated with milk fat and other milk traits. These genes were also found to be annotated for other milk traits such as milk yield, protein yield, and percentage.

Among all genes, 24 genes were found to be associated with both fat yield and fat percentage traits only (Appendix A). The common genes (125) were found enriched in biosynthetic-, catabolic-, regulatory-, transportation-, and cellular-response-associated metabolic processes. Among the metabolic genes associated with milk fat traits, the genes associated with milk fatty acid metabolism were *FASN, GPAT4, DGKG, ELOVL6*, and *LIPIN1* (Appendix A).

### 3.2. Milk Transcriptome Data Processing and Gene Expression Analysis of Fat QTL Genes

The publicly available RNA-seq bioproject (PRJNA419906) has library sizes ranging from 7764992200–13682843400 bp and 6842583600–12088807400 bp, for Jersey and Kashmiri, respectively (Appendix A). Further, gene expression counts per million (CPM), principal component analysis (PCA), and multidimensional scaling (MDS) (Appendix A) of the samples were assessed. The PCA and MDS plots of sequenced RNA-seq libraries showed a high level of similarity within breeds and relatively low variation between the lactation stages of breeds.

A total of 70 genes were found to be upregulated and 52 genes downregulated in the Jersey breed in comparison with the Kashmiri breed. Differentially expressed transcripts (DETs) were also explored for fat QTL genes (Appendix A). The volcano plots of differentially expressed genes (DEGs) and DETs depicting the distribution of upregulated and downregulated genes are shown in Figure 2.

In both the Jersey and Kashmiri breeds, Beta-lactoglobulin (LOC113901792), Casein beta (*CSN2*), and Casein alpha s1 (*CSN1S1*) were identified to be among the highly expressed top 20 genes (Appendix A). The DETs are listed in Appendix A. In the Jersey breed, *CXCL-8, TLR4,* and *OLR1* were among the highly upregulated genes. *CXCL-8/IL8,* produced by macrophages, epithelial cells, and airway smooth muscle cells, is a neutrophil chemotactic factor that induces chemotaxis in target cells and other granulocytes to initiate movement toward infection sites, whereas *OLR1*, a receptor on macrophages, epithelial cells, and airway smooth muscle cells, is involved in rapid oxidization of low-density lipoprotein (*LDL*), which is more readily recognized by the *TLR4* receptor. The list of top 10 upregulated and downregulated DEGs is given in Appendix A.

### 3.3. Protein Interaction Network Analysis of Milk Fat QTL Genes

A protein interaction network analysis was performed among 417 milk fat QTL genes, and the interaction network was generated with 403 nodes and 671 edges. Based on the degree of association, 50 hub and 50 bottleneck genes were selected from the network (Appendix A). A total of 74 QTL genes were found to be either hub or bottleneck genes (Figure 1B). Out of these, 25 genes (which accounted for 18 hubs/17 bottleneck genes) were differentially expressed in Jersey and Kashmiri (Figure 1C, Appendix A). Out of these genes, ten genes possessed both hub and bottleneck gene characteristics. The *SRC* and *DGAT1* genes were among the top differentially expressed hub and bottleneck genes (Table 1). *SRC* had the highest degree of association (30) with a log_2_ fold change of 1.480587, followed by *DGAT1* with a degree of 25 and a log_2_FC of 0.921104.

### 3.4. Variant Analysis of Hub and Bottleneck Genes among Indigenous Breeds

Illumina short read (Paired end) data of 14 samples from both groups of high- and-low-milk-yield breeds had 12.77 billion reads. After preprocessing, clean data included 11.02 billion reads, which is ca. 1516 Gb data. Each dataset had a minimum sequencing depth of ≥30x with an average GC content of 45.26%. The processed datasets contained on average 97.91% Q20 bases and 93.97% Q30 bases. The high-quality trimmed data aligned to the Brahman reference genome with an overall alignment rate of >95%. Initial variant calling on the aligned data provided 63,357,363 variants, which were filtered for high quality. After quality filtering on Q20, a total of 33,976,892 SNPs were identified across the genomes (Appendix A). Upon GATK analysis, 39,625,917 variants were found to pass the variant calibration. A total of 25,956,231 SNPs were found to be common among the variant callers. From these, SNPs in the 25 differentially expressed hub and bottleneck milk fat QTL genes were extracted, out of which 20 genes were found to have non-synonymous substitutions in the coding regions (Table 2).

The variants identified in these 20 genes were further explored for two kinds of genomic variant patterns, i.e., fixed SNP pattern in the cattle of the high-milk-yield group vs. variable SNP pattern in cattle of the low-milk-yield group, or vice versa. The fixed SNP pattern in high-milk-yield breeds in comparison to low-milk-yield breeds was observed in the genes *GHR, TLR4, LPIN1, CACNA1C, ZBTB16, ITGA1, ANK1*, and *NTG5E* (Table 3), and the opposite was observed in the genes *MFGE8, FGF2, TLR4, LPIN1, NUP98, PTK2, ZTB16, DDIT3*, and *NT5E* (Table 4). SNPs C/G, C/A, and G/A were confirmed in *GHR, TLR4*, and *LPIN1* in the Amritmahal, Pulikulam, and Dangi breeds (low-milk-yield) as against SNPs C/C, C/C, and G/G in the Gir and Sahiwal breed (high-milk-yield), respectively (Figure 3) (Appendix A). In the *TLR4* gene, variant g.107083326A>C was found in the low-milk-yield group, but the same variant was fixed in the high-milk-yield group. Similarly, in the *LPIN1* gene variant, g.85211528C>G was found in the low-milk-yield group, but this variant was fixed in the high-milk-yield group. In the *NUP98* gene and *LPIN1* variants, g.32707374G>A and g.85205642T>G, respectively, were found in the high-milk-yield group, but were found to be fixed (g.32707374G>G; g.85205642T>T) in the low-milk-yield group. Another *LPIN1* variant, g.85205642T>G, was observed in the high-milk-yield group, but was fixed in the low-milk-yield group. *LPIN1* and *ITGA1* had the maximum SNP count of 10, and the genes *PTK2, IGF1R, DDIT3, CXCL8*, and *LPL* had the least SNP count (Table 2).

## 4. Discussion

The *Bos indicus* genome is an interesting model to study the genomic potential of different indigenous cattle breeds such as Sahiwal, Gir, Amritmahal, Dangi, Gaolao, Deoni, Pulikulam, and Hallikar, which are highly adapted to different tropical conditions with varying milking potential. The availability of bovine QTL resources such as the Animal QTL database and the collection of QTLs for different traits have provided the opportunity to investigate genomic variation among indigenous breeds for milk-associated traits. Milk quality such as fat yield and percentage are highly variable traits among breeds. Jersey is one among the high-milk-producing breeds worldwide, whereas Kashmiri is one of the poorly performing breeds in the Kashmir region of India. Therefore, we aimed at differences in the expression of fat QTL genes between the two contrasting breeds, Jersey and Kashmiri. In this study, significantly expressed hub and bottleneck fat QTL genes were further analyzed to identify the genomic variants from the whole-genome sequence data between high- (Sahiwal and Gir) and low- (Amritmahal, Dangi, Gaolao, Deoni, Pulikulam, and Hallikar) milk-yield indigenous breeds. This understanding of low- and high-milk-yield breeds for milk fat quality may help in enhancing the quality of milk in the long run.

To explore the fat QTL genes, MEC RNA-seq data were processed and analyzed. The high level of similarity within breeds and relatively low variation between lactation confirmed the selection of RNA-seq data to explore differences between breeds rather than to explore difference in lactation stages of breeds. Among the highly expressed genes identified, it was observed that the Jersey breed has allocated more resources for the immune system, whereas the Kashmiri breed for regulation of ribosomal proteins. Among the top 10 upregulated genes, *CXC* motif chemokine ligand 8 was the most-upregulated gene. It is reported to be involved in various biological pathways such as increased insulin resistance, uncoupling of the GH/IGF1 axis, and an increase in mammary cell proliferation to improve metabolic health and milk yield [43]. Diacylglycerol kinase gamma (*DGKG*), another upregulated gene, is a member of the type I diacylglycerol kinases and is highly upregulated (log_2_FC = 4.03) in Jersey. It plays a role in lipid metabolism by modulating the balance between diacylglycerol and phosphatidic acid. Phosphatidic acid is a lipid second messenger to activate protein kinase C isoforms, ras guanyl nucleotide-releasing proteins, and some transient receptor potential channels [44]. Most of the top-upregulated genes in Jersey have been found to have a role in adipogenesis (*ETS2*) [45], adipocyte differentiation (*OLR1, PARM1*) [46,47], glucose transport (*SLC6A9*) (log_2_FC = 4.49) [48], glucose uptake (*SLC45A4*) [49], thyroid hormone synthesis (*TG*) [50], and aldosterone secretion (*KCNK9*) [51]. The upregulation of these genes in Jersey indicates their involvement in lipid biosynthesis in the mammary gland during lactation. Furthermore, UDP-glucose 6-dehydrogenase (*UGDH*), which is involved in the biosynthesis of glycosaminoglycans, hyaluronan, chondroitin sulfate, and heparan sulfate, has been found (log_2_FC = 0.75) to be upregulated in the Jersey breed. *UGDH*’s expression pattern in liver cells has been associated with an indispensable role in the metabolism of carbohydrates, fats, and proteins in dairy cattle [52]. Moreover, *UGDH* has been found close to two reported QTLs for fat yield, fat percentage, and protein yield [53].

During lactation, various morphological changes happen in the mammary tissue to support cellular differentiation, tissue elasticity, and reduced fat storage capacity in the animal. Upregulated *GRH13* is a transcription factor that mediates the proliferation of epithelial cells [54]. Similarly, *MTMR3/3-PAP*, a catalytically inactive member of the myotubularin gene family that coprecipitates the activity of lipid phosphatidylinositol 3-phosphate-3-phosphatase, is upregulated [55] in Jersey. Matrilin 2 (*MATN2*) and prolyl 4 hydroxylase (*P4HA3*), which are important to maintaining the newly synthesized collagen’s stability, are also upregulated [56,57]. *P4HA3* catalyzes the formation of 4-hydroxyproline (Hyp), which ensures the proper folding of procollagens during post-translational modification. The upregulation of *MATN2* and *P4HA3* probably may help in increasing the elasticity of the udder gland in Jersey during lactation. Further, the downregulation of *MFGE8* and *ELOVL16* may be responsible for the high fat content in milk and the shift to C18 from C16 fatty acids, respectively, in Jersey. *MFGE8* regulates the absorption of free fatty acids and increases intracellular triglycerides’ hydrolase activity, thereby restricting the storage of fat [58], whereas ELOVL fatty acid elongase (*ELOVL16*) elongates C16 saturated and monounsaturated fatty acids to C18 fatty acids [59]. Further, *IDH1*, which catalyzes the conversion of isocitrate to α-ketoglutarate and generates the primary source of NADPH for de novo fatty acid synthesis [60], has been to be found downregulated in Jersey (log_2_FC = −1.632). The dysregulation of the genes *MFGE8, EVOLVL16*, and *IDH1* might be responsible for the variation in the fat yield and composition in Jersey and Kashmiri cows. In addition, the decreased expression of these genes may be linked to the increased expression of genes involved in metabolism, glucose transport, and other transport activities, leading to higher milk production performance in Jersey.

Among the ten QTLdb milk and milk trait genes that were differentially expressed and had hub and bottleneck gene characteristics, four genes were found enriched in metabolic pathways (Appendix A). *SRC* was identified as the top hub gene interacting with several genes in the protein–protein interaction network. *SRC*, a non-receptor tyrosine kinase, performs a wide variety of cellular functions in terms of metabolism and is primarily involved in impaired glucose uptake [61]. Genes Diacylglycerol O-acyltransferase 1 (*DGAT1*) and ecto-5′-nucleotidase (*NT5E*) possess hub and bottleneck gene features. *DGAT1* encodes a protein that catalyzes the conversion of diacylglycerol and fatty acyl CoA to triacylglycerol. *DGAT1* is one of the highly studied genes for milk yield and fat quality [62,63]. The *NT5E* gene encodes a plasma membrane protein that catalyzes the conversion of extracellular nucleotides to membrane-permeable nucleosides. *SRC* and *DGAT1* were found to be highly upregulated in Jersey with log_2_FC = 1.48 and log_2_FC = 0.92, respectively. In the network, *DGAT1* was found to interact with Glycerol-3-phosphate acyltransferase 4 (*GPAT4*) (also known as *AGPAT6)*. This gene has been found to be involved in triglyceride biosynthesis and is comprised of acyltransferase motifs, which are essential for binding to substrates and catalyzing acyltransferase reactions. The *AGPAT6* gene, which is highly expressed in mammary gland epithelium during lactation [64], was also observed to be upregulated. *NT5E* was found to interact with *BDNF, NTRK2*, and *ZNRF4*. *NT5E* is involved in various biological process such as adenosine biosynthetic process, AMP catabolic process, leukocyte cell–cell adhesion, and negative regulation of inflammatory response [65]. *DGKG,* which was upregulated, was found to be a bottleneck gene in the network and had interactions with *PRKCG* and *RNT*. Phosphatidate phosphatase (*LPIN1*), an enzyme involved in lipid metabolism [66], was upregulated and found as a hub gene in the network having interactions with *PPARA* and *PPARGC1A*. *LPIN1* is involved in various biological processes related to lipid metabolism such as the triglyceride biosynthetic process, the fatty acid catabolic process, and the regulation of transcription by RNA polymerase II [66]. The gene *PTK2* is well known for its association with milk production traits [67], and *CACNA1A* has a role in hormone regulation of lactation [68]. Similarly, *ZBTB16* is involved in bovine adipogenesis [69]. *NUP98* is associated with protein percentage [70]. *TLR4* is a mastitis-associated marker [71]. *FGF2* expression is reported to be associated with milk production traits [72]. In this study, breed (high-milk-yield and low-milk-yield groups) differences in nsSNPs within these hub and bottleneck genes (*GHR, TLR4, LPIN1, CACNA1C, MFGE8, PTK2, ZBTB16, FGF2*, and *NUP98*) were identified. These fat QTL nsSNPs may have a role in the existing fat and milk yield differences between the breed groups. However, further studies to evaluate the impact of these SNPs on fat yield/percentage or milk yield need to be carried out.

## 5. Conclusions

In this study, initially, DEGs in Jersey epithelial cells were identified, and these were further explored in the QTL database as being the major hub and bottleneck genes. The transcriptome in Jersey indicated higher expression of genes involved in metabolism, glucose transport, and other transport activities, leading to higher milk production performance. The 20 differentially expressed hub and bottleneck fat QTL genes were explored for non-synonymous genomic variants in the whole-genome sequence data, which were generated from fourteen animals. The fixed SNP pattern in high-milk-yield breeds in comparison to low-milk-yield breeds was observed in the genes *GHR, TLR4, LPIN1, CACNA1C, ZBTB16, ITGA1, ANK1*, and *NTG5E*, and the opposite was observed in the genes *MFGE8, FGF2, TLR4, LPIN1, NUP98, PTK2, ZTB16, DDIT3*, and *NT5E*. The role of these SNPs needs to be further explored.

## Figures and Tables

**Figure 1 animals-13-00884-f001:**
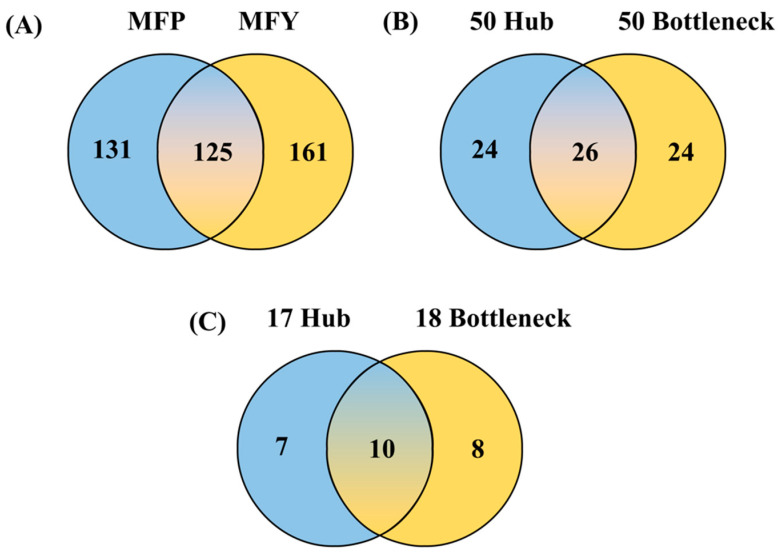
Venn diagram showing the hub and bottleneck genes: (**A**) QTL genes of milk fat percentage (256) and milk fat yield (286), (**B**) genes of top 50 hub and bottleneck genes, (**C**) genes of 17 hub and 18 bottleneck differentially expressed genes. MFP = milk fat percentage, MFY = milk fat yield.

**Figure 2 animals-13-00884-f002:**
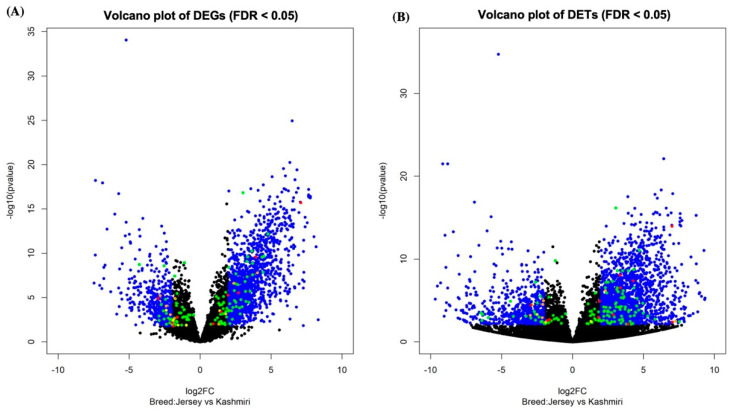
Differentially expressed milk-trait-associated QTL genes and transcripts in the milk transcriptome of Jersey and Kashmiri breeds (**A**) DEGs and (**B**) DETs. Volcano plot showing the significance versus the fold change. Blue dots (*p* < 0.05 and log_2_FC > 2): all *Bos indicus* genes; green dots (*p* < 0.05): QTL genes; orange dots (*p* < 0.05): bottleneck genes; yellow dots (*p* < 0.05): hub genes; red dots (*p* < 0.05): genes possessing hub and bottleneck features.

**Figure 3 animals-13-00884-f003:**
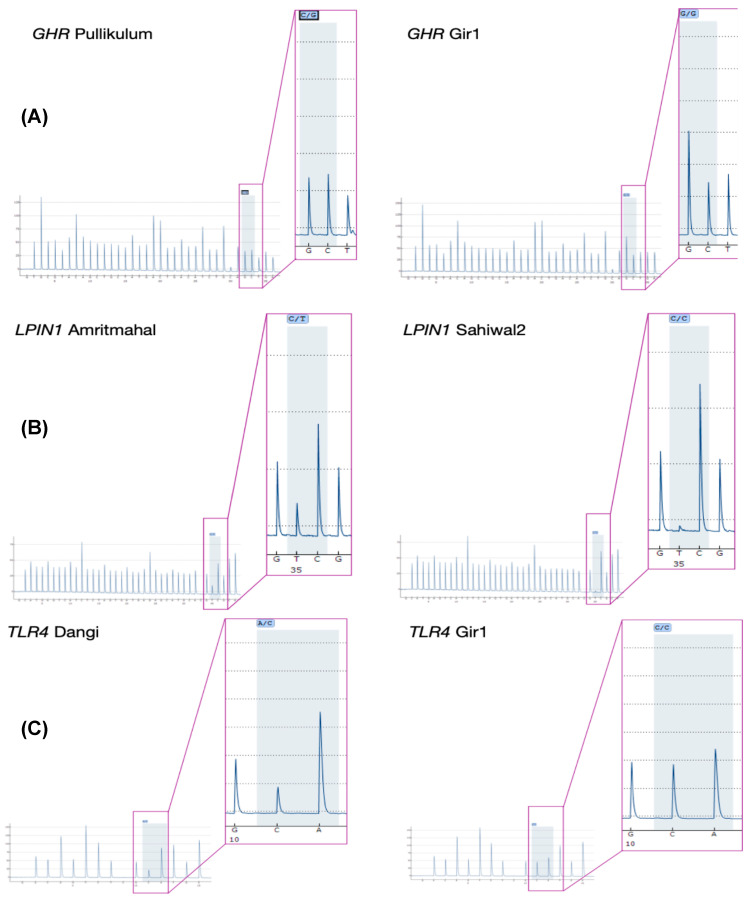
Confirmation of mutations shown in pyrograms with PyroMark SNP analysis assays: (**A**) *GHR* Gy392Ala (C>G) in Pulikulam (low-milk-yield breed) and (C/C) in Gir (high-milk-yield breed); (**B**) *LPIN1* a Arg772Lys (G>A) in Amritmahal (low-milk-yield breed) and (G/G) in Sahiwal (high-milk-yield breed); (**C**) *TLR4* a Asn347Gly (A>C) in Dangi (low-milk-yield breed) and (C/C) in Gir (high-milk-yield breed); the pyroseq sequencing primer is in reverse.

**Table 1 animals-13-00884-t001:** Differentially expressed hub and bottleneck QTL genes of the protein interaction network of fat QTL genes.

Gene ID	log_2_FC	*p*-Value	Trait Name	Description
Hub genes				
ENSBTAG00000008938 (*SRC*)	1.480587	0.002561	MFY; MPP; MPY; TDMY	*SRC* proto-oncogene, non-receptor tyrosine kinase
ENSBTAG00000026356 (*DGAT1*)	0.921104	0.032625	MFP; MFY; MPP. MPY; MKCP; MY	Diacylglycerol O-acyltransferase 1
ENSBTAG00000007867 (*STAT1*)	1.938208	0.0000648	MFP; MPP; MPY; MY	Signal transducer and activator of transcription 1
ENSBTAG00000005691 (*FGF2*)	−1.55871	0.042228	MFY; MY	Fibroblast growth factor 2
ENSBTAG00000019716 (*CXCL8*)	7.097327	0.000000000000077	MFY; MPP; MPY;TDMY	C-X-C motif chemokine ligand 8
ENSBTAG00000006240 (*TLR4*)	3.918383	0.0000000123	MFP; MPP; MY	Toll like receptor 4
ENSBTAG00000001335 (*GHR*)	−1.97171	0.02566	MFP; MFY; MPP; MPY; MY	Growth hormone receptor
ENSBTAG00000012855 (*LPL*)	−1.61377	0.008224	MFP; MFY; MPP	Lipoprotein lipase
ENSBTAG00000009578 (*PTK2*)	−0.97617	0.043672	MFP; MFY; MPP; MPY; MY	Protein tyrosine kinase 2
ENSBTAG00000000546 (*ERBB2*)	−1.89535	0.048867	MFP; MFY; MPP; MPY; MY	Erb-b2 receptor tyrosine kinase 2
ENSBTAG00000021527 (*IGF1R*)	1.798566	0.003947	MFP; MFY; MPP; MPY; MY	Insulin like growth factor 1 receptor
ENSBTAG00000007476 (*BTRC*)	2.698919	0.0000384	MFY	Beta-transducin repeat containing E3 ubiquitin protein ligase
ENSBTAG00000014357 (*SDC2*)	−1.96312	0.029226	MFP; MPP	Syndecan 2
ENSBTAG00000048655 (*NT5E*)	−2.97069	0.000136	MFP; MFY; MPY	5′-nucleotidase ecto
ENSBTAG00000007689 (*LPIN1*)	−2.38799	0.001822	MFP; MFY; MPP; MPY; TDMY	Lipin 1
ENSBTAG00000003300 (*MFGE8*)	−1.90426	0.015698	MFP; MPP	Milk fat globule EGF and factor V/VIII domain containing
ENSBTAG00000020536 (*HERC6*)	1.37883	0.00206	MFP; MFY; MPP; MPY	HECT and RLD domain containing E3 ubiquitin protein ligase family member 6
Bottleneck Genes				
ENSBTAG00000026356 (*DGAT1*)	0.921104	0.032625	MFP; MFY; MPP; MPY; MKCP; MY	Diacylglycerol O-acyltransferase 1
ENSBTAG00000008938 (*SRC*)	1.480587	0.002561	MFY; MPP; MPY; TDMY	*SRC* proto-oncogene, non-receptor tyrosine kinase
ENSBTAG00000008432 (*NUP98*)	1.307627	0.033165	MFP	Nucleoporin 98 and 96 precursor
ENSBTAG00000027629 (*ANK1*)	−2.3949	0.000752	MFY; MPY	Ankyrin 1
ENSBTAG00000011266 (*ZBTB16*)	−1.84946	0.011504	MFY	Zinc finger and BTB domain containing 16
ENSBTAG00000016525 (*ITGA1*)	−2.10583	0.004815	MFY; MPP; MPY; MY	Integrin subunit alpha 1
ENSBTAG00000019716 (*CXCL8*)	7.097327	0.000000000000077	MFY; MPP; MPY; TDMY	C-X-C motif chemokine ligand 8
ENSBTAG00000007867 (*STAT1*)	1.938208	0.0000648	MFP; MPP; MPY; MY	Signal transducer and activator of transcription 1
ENSBTAG00000000546 (*ERBB2*)	−1.89535	0.048867	MFP; MFY; MPP; MPY; MY	Erb-b2 receptor tyrosine kinase 2
ENSBTAG00000007476 (*BTRC*)	2.698919	0.0000384	MFY	Beta-transducin repeat containing E3 ubiquitin protein ligase
ENSBTAG00000010106 (*CCND3*)	−1.78227	0.000135	MFP; MPP; MPY; MY	Cyclin D3
ENSBTAG00000001335 (*GHR*)	−1.97171	0.02566	MFP; MFY; MPP; MPY; MY	Growth hormone receptor
ENSBTAG00000006240 (*TLR4*)	3.918383	0.0000000123	MFP; MPP; MY	Toll like receptor 4
ENSBTAG00000010660 (*CACNA1C*)	−2.82445	0.0000723	MFY	Calcium voltage-gated channel subunit alpha1 C
ENSBTAG00000012855 (*LPL*)	−1.61377	0.008224	MFP; MFY; MPP	Lipoprotein lipase
ENSBTAG00000031544 (*DDIT3*)	2.367342	0.0000047	MFP; MFY; MPP; MPY; MY	DNA damage inducible transcript 3
ENSBTAG00000005091 (*DGKG*)	4.032464	0.000000442	MFP; MPP; MY	Diacylglycerol kinase gamma
ENSBTAG00000048655 (*NT5E*)	−2.97069	0.000136	MFP; MFY; MPY	5′-nucleotidase ecto

MFP, milk fat percentage; MFY, milk fat yield; MPP, milk protein percentage; MPY, milk protein yield; MKCP, milk kappa casein protein; MY, milk yield.

**Table 2 animals-13-00884-t002:** Summary of variant analysis of differentially expressed hub and bottleneck genes.

Gene ID	Chr	Start	End	SNP Count	BiSNP
ENSBTAG00000007689 (*LPIN1*)	NC_040086.1	85168990	85299051	10	9
ENSBTAG00000009578 (*PTK2*)	NC_040089.1	2793205	2986179	1	1
ENSBTAG00000026356 (*DGAT1*)	NC_040089.1	547336	558673	2	2
ENSBTAG00000014357 (*SDC2*)	NC_040089.1	67885139	68006993	2	2
ENSBTAG00000008432 (*NUP98*)	NC_040090.1	32662235	32749000	2	2
ENSBTAG00000011266 (*ZBTB16*)	NC_040090.1	59519169	59723887	3	3
ENSBTAG00000005691 (*FGF2*)	NC_040092.1	37900967	37980582	2	2
ENSBTAG00000005091 (*DGKG*)	NC_040076.1	80595457	80821976	3	3
ENSBTAG00000016525 (*ITGA1*)	NC_040095.1	25934013	26116012	10	9
ENSBTAG00000001335 (*GHR*)	NC_040095.1	31683009	31993386	6	6
ENSBTAG00000003300 (*MFGE8*)	NC_040096.1	20516116	20540642	4	4
ENSBTAG00000021527 (*IGF1R*)	NC_040096.1	7856996	8161856	1	1
ENSBTAG00000027629 (*ANK1*)	NC_040102.1	36035255	36276595	5	4
ENSBTAG00000010660 (*CACNA1C*)	NC_040080.1	11268673	11659827	3	3
ENSBTAG00000031544 (*DDIT3*)	NC_040080.1	64346323	64350540	1	1
ENSBTAG00000020536 (*HERC6*)	NC_040081.1	36492384	36549974	9	9
ENSBTAG00000019716 (*CXCL8*)	NC_040081.1	88364933	88368713	1	1
ENSBTAG00000006240 (*TLR4*)	NC_040083.1	107075099	107086126	6	5
ENSBTAG00000012855 (*LPL*)	NC_040083.1	66717576	66744131	1	1
ENSBTAG00000048655 (*NT5E*)	NC_040084.1	64029657	64102659	6	6

BiSNP, biallelic SNP.

**Table 3 animals-13-00884-t003:** Genes with a fixed SNP pattern in high-milk-yield breeds (Sahiwal and Gir) and the respective variable SNP pattern in low-milk-yield breeds (Gaolao, Deoni, Pulikulam, Hallikar, Dangi, and Amritmahal).

Gene		*GHR*	*GHR*	*TLR4*	*TLR4*	*LPIN1*	*LPIN1*	*LPIN1*	*CACNA1C*	*ZBTB16*	*ITGA1*	*ANK1*	*ANK1*	*NT5E*	*NT5E*
Genomic location		31685773	31685984	107083326	107083914	85187074	85209309	85211528	11271411	59717709	25983121	36054194	36076037	64035090	64065719
Genomic Variant	Ref	C	A	A	C	G	C	C	C	C	C	G	G	C	G
	Alt	G	G	C	A	A	T	G	T	T	T	A	A	T	A
Protein Variant Pos(Ref/Alt)		392 (G/A)	462 (N/D)	151 (A/T)	347 (N/G)	772 (R/K)	631 (A/E)	542 (R/P)	204 (E/K)	393 (V/M)	588 (V/M)	111 (P/S)	127 (R/H)	475 (C/F)	151 (A/V)
Sahiwal 1		C/C	A/A	A/A	C/C	G/G	C/C	C/C	C/C	C/C	C/C	G/G	G/G	C/C	G/G
Sahiwal 2		C/C	A/A	A/A	C/C	G/G	C/C	C/C	C/C	C/C	C/C	G/G	G/G	C/C	G/G
Sahiwal 3		C/C	A/A	A/A	C/C	G/G	C/C	C/C	C/C	C/C	C/C	G/G	G/G	C/C	G/G
Sahiwal 4		C/C	A/A	A/A	C/C	G/G	C/C	C/C	C/C	C/C	C/C	G/G	G/G	C/C	G/G
Gir 1		C/C	A/A	A/A	C/C	G/G	C/C	C/C	C/C	C/C	C/C	G/G	G/G	C/C	G/G
Gir 2		C/C	A/A	A/A	C/C	G/G	C/C	C/C	C/C	C/C	C/C	G/G	G/G	C/C	G/G
Gir 3		C/C	A/A	A/A	C/C	G/G	C/C	C/C	C/C	C/C	C/C	G/G	G/G	C/C	G/G
Gir 4		C/C	A/A	A/A	C/C	G/G	C/C	C/C	C/C	C/C	C/C	G/G	G/G	C/C	G/G
Amritmahal		C/C	A/A	A/A	C/C	G/A	C/C	C/C	C/C	C/C	C/C	G/G	G/G	C/C	G/G
Dangi		C/G	A/G	A/C	C/A	G/G	C/C	C/G	C/C	C/T	C/C	G/G	G/G	C/C	G/G
Gaolao		C/G	A/A	A/C	C/C	G/G	C/T	C/C	C/C	C/C	C/C	G/G	G/G	T/T	G/G
Deoni		C/C	A/A	A/C	C/C	G/G	C/C	C/G	C/T	C/T	C/C	G/G	G/G	C/C	G/A
Pulikulam		C/G	A/A	A/A	C/C	G/A	C/C	C/T	C/C	C/C	C/T	G/A	G/G	C/C	G/G
Hallikar		C/C	A/A	A/A	C/C	G/A	C/C	C/G	C/C	C/C	C/C	G/G	G/A	C/C	G/G

**Table 4 animals-13-00884-t004:** Genes with fixed SNP pattern in low-milk-yield breeds (Sahiwal and Gir) and respective variable SNP pattern in high-milk-yield breeds (Gaolao, Deoni, Publicum, Hallikar, Dangi and Amritmahal).

Gene		*MFGE8*	*FGF2*	*TLR4*	*LPIN1*	*NUP98*	*PTK2*	*ZBTB16*	*ZBTB16*	*DDIT3*	*NT5E*
Genomic location		20518217	37920746	107080326	85205642	32707374	2973942	59717979	59718206	64346576	64102580
Genomic Variant	Ref	A	G	G	T	G	A	T	G	C	G
	Alt	T	A	A	G	A	C	C	A	T	T
Protein Variant Pos(Ref/Alt)		328 (S/R)	19 (G/R)	67 (R/K)	766 (S/P)	548 (H/Y)	904 (D/A)	598 (G/R)	627 (A/V)	87 (S/L)	8 (T/N)
Sahiwal 1		A/A	G/G	G/A	T/T	G/G	A/A	T/T	G/G	C/T	G/G
Sahiwal 2		A/A	G/G	G/G	T/G	G/G	A/A	T/T	G/G	C/C	G/T
Sahiwal 3		A/A	G/A	G/G	T/T	G/A	A/A	T/T	G/A	C/C	G/G
Sahiwal 4		A/A	-	G/G	T/T	G/A	A/A	T/T	G/G	C/C	G/G
Gir 1		A/T	-	G/G	T/T	G/G	A/A	T/T	G/G	C/C	G/G
Gir 2		A/A	-	G/G	T/T	G/G	A/A	T/T	G/G	C/C	G/G
Gir 3		A/A	G/G	G/G	T/T	G/G	A/C	T/C	G/G	C/C	G/G
Gir 4		A/A	G/G	G/G	T/T	G/G	A/A	T/T	G/G	C/C	G/G
Amritmahal		A/A	G/G	G/G	T/T	G/G	A/A	T/T	G/G	C/C	G/G
Dangi		A/A	G/G	G/G	T/T	G/G	A/A	T/T	G/G	C/C	G/G
Gaolao		A/A	G/G	G/G	T/T	G/G	A/A	T/T	G/G	C/C	G/G
Deoni		A/A	G/G	G/G	T/T	G/G	A/A	T/T	G/G	C/C	G/G
Pulikulam		A/A	G/G	G/G	T/T	G/G	A/A	T/T	G/G	C/C	G/G
Hallikar		A/A	G/G	G/G	T/T	G/G	A/A	T/T	A/A	C/C	G/G

## Data Availability

All the used RNA-seq data are available at NCBI SRA under the project accession number PRJNA419906. Genomics for conservation of indigenous cattle breeds and for enhancing milk yield BT/PR26466/AAQ/1/704/2017.

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
