# Peer review of "Non-Synonymous Variants in Fat QTL Genes among High- and Low-Milk-Yielding Indigenous Breeds"

_animals, 2023, doi:10.3390/ani13050884_

Round 1

Reviewer 1 Report

The paper treats a very important subject.

However, the objective was not cited.

methodology can be imporoved

Author Response

We are grateful to the reviewer for their feedback on the manuscript. Below we answer their queries and incorporate suggestions in the revised version of the manuscript.

Thanks for your valuable remark

Comment 1: The paper treats a very important subject. However, the objective was not cited. methodology can be improved.

Response. The objective of the study is to see the variation across indigenous breeds for milk fat QTL. The methodology is improved after suggestions.

Reviewer 2 Report

1. First, the format of the article must be major revised.

2. In Figure 3, the results of PCA and MDS show that individuals of the different groups cannot get together. Is the analysis in this article statistically significant?

Author Response

We are grateful to the reviewer for their feedback on the manuscript. Below we answer their queries and incorporate suggestions in the revised version of the manuscript.

Thanks for your valuable remark

Comment 1: First, the format of the article must be major revised.

Response. The format of the article is revised.

Comment 2: In Figure 3, the results of PCA and MDS show that individuals of the different groups cannot get together. Is the analysis in this article statistically significant?

Response. As suggested by other reviewers we are shifting figure 3 to supplementary.

Reviewer 3 Report

The author used the bioinformatics strategy to mine some variants associated with milk

fat traits in cattle. This is an interesting work. The author provide a new idea how to find some variants associated with traits of interest by intergrating omic data. Nevertheless, some questions need to be concerned.

1.The format and language of the manuscript still need to be further improved

2. The author is not very concerned about the sample selection. In the study, the description of the breed information is too simplistic, the author should provide more detailed sample information.

3.Line 33. please provide the detail information on the crossbred animals.

4.Line 84. where is the list of QTL genes?

5.Line 85. what functional annotation was used in the present study?

6.Line 86. please cite the reference.

7.Line 96. please provide the proof to support your idea. Why Jersey is a high-fat content breed? Why Kashmiri is a low-fat content breed? As I know, these samples from different lactation, So, I'm very skeptical about your handling.

8.Line 128-130. please provide the detail parameters.

Author Response

We are grateful to the reviewer for their feedback on the manuscript. Below we answer their queries and incorporate suggestions in the revised version of the manuscript.

Thanks for your valuable remark

Comment 1: The format and language of the manuscript still need to be further improved

Response. The format and language is corrected as per suggestion.

Comment 2: The author is not very concerned about the sample selection. In the study, the description of the breed information is too simplistic, the author should provide more detailed sample information.

Response. We have taken data from bioproject PRJNA419906. The samples were transcriptome of mammary epithelial cells milk representing early lactation, mid-lactation and late lactation stages, respectively. We have added the details as suggested.

Comment 3: Line 33. please provide the detail information on the crossbred animals.

Response. The crossbred animal population is corresponding to 28% of percentage share of milk production during 2020-21. To enhance the milk production in country, earlier several schemes were implemented, emphasis on crossbreeding. With milk productivity the number of crossbred cattle also increased. Since our objective have only focussed on indigenous cattle breeds, we haven’t discussed crossbred cattle in detail. The reference we have added which is having all details of cross-bred animals.

Comment 4: Line 84. where is the list of QTL genes?

Response. The list is provided in supplementary material 1. In the supplementary, 286 gene correspond to milk fat yield and 256 gene correspond to milk fat percentage. There is 125 common genes between them so altogether 417 genes for fat trait is present.

Comment 5: Line 85. what functional annotation was used in the present study?

Response. The functional annotation is done for the terms related to biological processes.

Comment 6: Line 86. please cite the reference.

Response. Citation is added.

Comment 7: Line 96. please provide the proof to support your idea. Why Jersey is a high-fat content breed? Why Kashmiri is a low-fat content breed? As I know, these samples from different lactation, So, I'm very skeptical about your handling.

Response. The fat content of Jersey ranged from 4.10% - 4.86%, corresponding to Kashmiri 3.20% -3.94%. They recorded ±7 days relative to the day of sampling and quantified via Milk-autoanalyzer. Here is the attached research article which support the fat content of Jersey and Kashmiri https://doi.org/10.1371/journal.pone.0211773.

Comment 8: Line 128-130. please provide the detail parameters.

Response. The detail parameters of variant calling is added in the methodology as suggested.

Reviewer 4 Report

Dear authors,

I will recommend your manuscript for publication, but I has some questions for edition.

L3 – fix title

L151 – scheme of workflow is not necessary and it is not result

L160 – wrong proportion in diagram. Crossed area must be bigger.

L172 – move parameters of samples to suppl. It is not result.

L185 – Are you performed expression analysis? I did not see it in Methods.

Generally – in Results too many tables with non-final results. Rewrite this part and move half of them to suppl.

Regards,

Author Response

We are grateful to the reviewer for their feedback on the manuscript. Below we answer their queries and incorporate suggestions in the revised version of the manuscript.

Thanks for your valuable remark

Comment 1: L3 – fix title

Response. We have fixed the title. Actually it was a typological error while uploading the manuscript.

Comment 2: L151 – scheme of workflow is not necessary and it is not result.

Response.  The workflow is shifted after introduction. It may help viewers to understand our work easily.

Comment 3: L160 – wrong proportion in diagram. Crossed area must be bigger.

Response.  Changes are made as suggested.

Comment 4: L172 – move parameters of samples to suppl. It is not result.

Response. As suggested Fig 3. is moved to supplementary

Comment 5: L185 – Are you performed expression analysis? I did not see it in Methods.

Response. We calculated the maximum expression of top genes transcriptome data mentioned in table 1. We didn’t performed gene expression analysis  in-vitro.

Comment 5: Generally – in Results too many tables with non-final results. Rewrite this part and move half of them to suppl.

Response. We have shifted the non-final tables 1-3 to supplementary considering your suggestions.

Round 2

Reviewer 3 Report

The revised manuscript has been greatly improved. However, these problem still needs to be solved:

1. Line 101-102, the author should defined the parameters for Cytohubba. what threshold value can be served as the significant degree of association.

2.  Line 110-112, the author has answered my comments. But I suggested that the author should provide this information into the main text.

3. Line 149, Also,  the author deal with the comment. please provide the detail parameters for QD, MQRankSum,.......

Author Response

We are appreciative of the reviewer's comments on the manuscript. We respond to their questions and consider their opinions in the updated manuscript version below.

Thanks for your valuable remarks

Comment 1:  Line 101-102, the author should defined the parameters for Cytohubba. what threshold value can be served as the significant degree of association.

Response. The threshold parameter which cytohubba plugin generally considers is added in the methodology.

Comment 2: Line 110-112, the author has answered my comments. But I suggested that the author should provide this information into the main text.

Response. Added the information in methodology section as suggested.

Comment 3:  Line 149, Also,  the author deal with the comment. please provide the detail parameters for QD, MQRankSum,.......

Response. The detail parameters were provided in the concerned section as suggested by reviewer.